# OpenReview forum: "Alignment of Large Language Models with Constrained Learning"
_NeurIPS.cc/2025/Conference — NeurIPS 2025 poster_

### Official Review · Reviewer_bN26 · 2025-07-01

**Clarity:** 3
**Significance:** 3
**Originality:** 3
**Rating:** 4
**Confidence:** 2

**Summary:**

This paper focuses on the alignment of large language models (LLMs) by incorporating constraints to manage conflicting rewards, such as helpful and harmful rewards. The authors propose a novel approach to address the alignment problem by imposing constraints on LLMs, which is crucial for ensuring that models adhere to specified reward constraints. The paper provides both theoretical and experimental contributions, with the abstract and introduction accurately summarizing these contributions and the scope of the work.

**Questions:**

See the weaknesses part.

**Ethical Concerns:**

["NO or VERY MINOR ethics concerns only"]

**Final Justification:**

I have read all the rebuttal and this is my final rating.

**Limitations:**

Yes

**Paper Formatting Concerns:**

No formatting concerns.

**Quality:**

3

**Strengths And Weaknesses:**

**Strength**

The work addresses a significant challenge in the alignment of LLMs with human values and constraints.

The paper presents a well-structured theoretical framework and supports it with experimental results.

The paper is clear in its presentation of the alignment problem and the proposed solution. The authors have discussed the broader impacts of their work, including potential societal benefits such as improved safety and fairness of LLMs.

**Weakness**

The experiments are limited to a 7b model and a single constraint. This may raise doubts about the generalizability of the results to various and larger models and more complex constraints.
How do the authors plan to extend their experiments to larger models and multiple constraints? Clarification on this could enhance the perceived impact of the work.

---

> ### Author Rebuttal · Authors · 2025-07-31
>
> ## Re: Weakness
>
> Our problem formulation and algorithm are not limited to single-constraint alignment tasks and are readily scalable to multi-constraint settings. We have designed and conducted additional experiments on a multi-constraint task using a new dataset with two distinct constraints (harmlessness and humor). Specifically, we fine-tune a LLaMA2-7B model on the Anthropic HH-RLHF dataset [1] to obtain a reference policy. We then align the model using our method under varying constraint levels. We use the helpfulness and harmlessness reward models from [6] for evaluating helpfulness and harmlessness, respectively, and the humor–no-humor model as the reward model for humor. We demonstrate that our iterative method satisfies the constraints while maximizing the primary utility on this task.
>
> For implementation, similar to our main experiment, we randomly select 1000 prompts from the full HH-RLHF dataset and sample 128 responses per prompt in the dual sub-gradient step. In the policy optimization step, we train the model using DPO with 10K preference pairs randomly selected from the same dataset. We evaluate our trained models on a randomly sampled set of 2000 prompts, each with 2 responses, from the test split of the HH-RLHF dataset.
>
> | Criterion | Expected improvement | Actual improvement      |
> |-----------|----------------------|-------------------------|
> | Harmless  | 0.5                  | 0.588 (-0.765 → -0.177) |
> | Humor     | 0.5                  | 0.907 (-0.110 → 0.798)  |
>
> The table above shows the results for threshold $\mathbf{b}=[0.5,0.5]$, i.e., an expected improvement of 0.5 in both harmlessness and humor scores. We observe that both constraints are satisfied, while the helpfulness score improves by 9.12%.
>
> We are happy to report additional results for other values of $\mathbf{b}$ during the discussion period and will include this multi-constraint task in the final version of our paper.
>
> Regarding the concern about using the 7B-scale model, we note that our theoretical results and algorithm are applicable to LLMs of any size. The 7B-scale models are sufficient to address the questions we focus on, namely constraint satisfaction and trade-offs among properties (see Section 4.2). Moreover, using 7B models is a common practice in the alignment literature [2,3,4,5], and adhering to this scale facilitates fair and direct comparison with prior work.
>
> We appreciate the reviewer's positive evaluation of our work, and believe that the additional experiments and discussions help address the concern regarding the scope of our experiments.
>
> **References:**
>
> [1] Training a Helpful and Harmless Assistant with Reinforcement Learning from Human Feedback
>
> [2] Safe RLHF: Safe Reinforcement Learning from Human Feedback
>
> [3] One-Shot Safety Alignment for Large Language Models via Optimal Dualization
>
> [4] Stepwise Alignment for Constrained Language Model Policy Optimization
>
> [5] Bounded Rationality for LLMs: Satisficing Alignment at Inference-Time
>
> [6] Rewards-in-Context: Multi-objective Alignment of Foundation Models with Dynamic Preference Adjustment

---

### Official Review · Reviewer_Qxpn · 2025-07-02

**Clarity:** 3
**Significance:** 2
**Originality:** 2
**Rating:** 4
**Confidence:** 3

**Summary:**

This paper studies the constrained alignment of LLMs, aiming to maximize primary objectives (e.g., helpfulness) while satisfying secondary constraints (e.g., safety). Building upon existing Lagrangian-based methods, the authors propose an iterative dual-based alignment approach (CAID) that alternates between LLM policy updates and dual variable updates. The paper provides:
- A formal theoretical analysis of the primal-dual gap between distribution space and LLM parameter space.
- Guarantees on the optimality gap of learned LLM policies under both objective and constraint functions.
- Experimental validation on the PKU-SafeRLHF dataset, showing improved constraint satisfaction and better trade-offs between helpfulness and safety compared to one-shot alignment baselines.

**Questions:**

-- Extend empirical evaluation to multi-constraint settings to demonstrate broader applicability.

-- Provide head-to-head comparisons with more recent constrained alignment techniques beyond one-shot alignment (e.g., Stepwise Alignment, constrained RLHF variants).

-- Include analysis of computational overhead and scalability for larger LLMs.

-- Broaden the experimental study to include additional LLM architectures and sizes to evaluate the robustness and generality of the proposed method.

**Ethical Concerns:**

["NO or VERY MINOR ethics concerns only"]

**Final Justification:**

The authors have conducted additional experiments to address my questions.

**Limitations:**

yes

**Quality:**

2

**Strengths And Weaknesses:**

**Strengths**

- The paper provides a careful theoretical analysis, including characterizing primal-dual gaps, quantifying optimality with respect to both objectives and constraints.
- Extends existing one-shot dual-based approaches to a more general multi-shot framework.
- Empirical validation on a public safety alignment benchmark (PKU-SafeRLHF), demonstrating improvements in constraint satisfaction and helpfulness-safety trade-offs.
- The paper is well-written.

**Weaknesses**
- The core algorithmic contribution (iterative dual-based alignment) is a relatively incremental extension of existing dual-based or one-shot methods like Huang et al. (NeurIPS 2024):
- Alternating Lagrangian maximization and dual updates is a well-known framework in constrained learning and optimization. The novelty of paper lies in formalizing its application to LLMs with theoretical guarantees, rather than proposing a fundamentally new alignment strategy.
- Empirical scope is limited:

  -- Only tested on a single model Alpaca-7B.

  -- Only single constraint experiments (safety), whereas multi-constraint setups (common in real-world alignment) are not explored.

  -- Comparisons are primarily against one-shot alignment; broader baselines (e.g., Step-Align, constrained RLHF variants) are absent.

---

> ### Author Rebuttal · Authors · 2025-07-31
>
> ## Re: Weakness 1 & 2
> We agree with the reviewer that alternating Lagrangian maximization and dual updates is a well-known framework in constrained optimization. Rather than proposing a new optimization framework, our contribution lies in demonstrating how to adapt and analyze this framework in the context of LLM alignment, resulting in an iterative, dualization-based alignment strategy that we believe is fundamental to constrained alignment problems. Importantly, the non-convexity of LLM parametrization introduces unique challenges in this development.
>
> Compared to Huang et al. (NeurIPS 2024), our method introduces two key algorithmic innovations. First, while Huang et al. (NeurIPS 2024) uses an exact optimal dual variable, which is hard to evaluate in practice (see it in Huang et al. (NeurIPS 2024)), our method removes this assumption and instead employs a dual-based descent procedure to approximately find an optimal dual variable in a tractable manner. This demonstrates our first algorithmic novelty. Second, motivated by our theoretical result showing that smaller optimal dual variables lead to smaller optimality gaps, we introduce warm-start initialization and post-processing techniques (see Algorithm 1 and Appendix D) to improve convergence performance. This demonstrates our second algorithmic novelty. These practical designs not only clearly generalize the method proposed in Huang et al. (NeurIPS 2024), but also significantly improve constraint satisfaction for practically aligned LLMs (see Figure 2).
>
> To the best of our knowledge, our work is the first to analyze the duality gap in solving constrained alignment problems and to provide theoretical guarantees on the suboptimality of practically aligned LLMs induced by dual iterations. These insights deepen our understanding of the trade-off between constraint satisfaction and performance when addressing constraints on pretrained LLMs. Our work fills a theoretical gap in the constrained alignment literature, including the work of Huang et al. (NeurIPS 2024), which focuses on simple algorithms without offering a duality and optimality analysis.
>
> ## Re: Weakness 3.1, 3.2 & Question 1
> Our problem formulation and algorithm are not limited to single-constraint alignment tasks and are readily scalable to multi-constraint settings. To address the concerns, we have designed and conducted additional experiments on a multi-constraint task using a new dataset with two distinct constraints (harmlessness and humor). Specifically, we fine-tune a LLaMA2-7B model on the Anthropic HH-RLHF dataset [1] to obtain a reference policy. We then align the model using our method under varying constraint levels. We use the helpfulness and harmlessness reward models from [6] for evaluating helpfulness and harmlessness, respectively, and the humor–no-humor model as the reward model for humor. We demonstrate that our iterative method satisfies the constraints while maximizing the primary utility on this task.
>
> For implementation, similar to our main experiment, we randomly select 1000 prompts from the full HH-RLHF dataset and sample 128 responses per prompt in the dual sub-gradient step. In the policy optimization step, we train the model using DPO with 10K preference pairs randomly selected from the same dataset. We evaluate our trained models on a randomly sampled set of 2000 prompts, each with 2 responses, from the test split of the HH-RLHF dataset.
>
> | Criterion | Expected improvement | Actual improvement      |
> |-----------|----------------------|-------------------------|
> | Harmless  | 0.5                  | 0.588 (-0.765 → -0.177) |
> | Humor     | 0.5                  | 0.907 (-0.110 → 0.798)  |
>
> The table above shows the results for threshold $\mathbf{b}=[0.5,0.5]$, i.e., an expected improvement of 0.5 in both harmlessness and humor scores. We observe that both constraints are satisfied, while the helpfulness score improves by 9.12%.
>
> We are happy to report additional results for other values of $\mathbf{b}$ during the discussion period and will include this multi-constraint task in the final version of our paper.
>
>
> ## Re: Weakness 3.3 & Question 2
> First, we would like to clarify some key differences between the recent constrained alignment techniques and our work. While many of them (such as Stepwise Alignment [4]) study trade-offs among multiple objectives or properties, our primary objective is to impose constraints on LLMs, i.e., to find an optimal *feasible* model. It is well-known that Pareto optimality in multi-objective optimization is fundamentally different from optimality under constraints. Additionally, the prior multi-objective alignment works typically do not address constraint satisfaction explicitly, as we do in Figure 2a.
>
> To compare trade-offs, we evaluate our method against the beaver-7b-v1.0 model from Safe-RLHF [2], as well as the best reported models, SACPO and P-SACPO, from Stepwise Alignment [4]. For evaluation, we use the GPT-4o-mini model as the judge and the same prompt dataset used in the GPT-4 evaluation from the Safe-RLHF paper. We observe that, at constraint levels $\mathcal{b}=8,9$, our method generally achieves higher win rates across both helpfulness and safety criteria (see the table below).
> |                | **$\mathcal{b}=8$**     |        | **$\mathcal{b}=9$**     |        |
> |----------------|-------------|--------|-------------|--------|
> |                | Helpfulness | Safety | Helpfulness | Safety |
> | beaver-7b-v1.0 | 0.58        | 0.49   | 0.6         | 0.54   |
> | sacpo          | 0.62        | 0.53   | 0.78        | 0.56   |
> | p-sacpo        | 0.54        | 0.51   | 0.52        | 0.52   |
>
> ## Re: Question 3, 4
> Please refer to Appendix D.2 for a detailed runtime comparisons of our main experiment. The computational cost of our method is comparable to that of the one-shot method, as each iteration requires only a single epoch of DPO. Improvements shown in Figure 2 are typically observed within four iterations. The additional runtime of the multi-shot method only comes from dual variable estimation, which can be efficiently accelerated through engineering techniques.
>
> Regarding the concern about using the 7B-scale model, we note that our theoretical results and algorithm are applicable to LLMs of any size. The 7B-scale models are sufficient to address the questions we focus on, namely constraint satisfaction and trade-offs among properties (see Section 4.2). Moreover, using 7B models is a common practice in the alignment literature [2,3,4,5], and adhering to this scale facilitates fair and direct comparison with prior work. For the robustness and generality aspects, we conduct evaluations using model-based methods (Section 4), GPT-based methods (Appendix E.2), and evaluate on an adversarial dataset (Appendix E.3). We will also include the additional multi-constraint experiment mentioned earlier in the final version of our paper.
>
> We appreciate the reviewer’s comments, and believe that the additional experiments and discussions provide further insight to open your evaluation of our work.
>
> **References:**
>
> [1] Training a Helpful and Harmless Assistant with Reinforcement Learning from Human Feedback
>
> [2] Safe RLHF: Safe Reinforcement Learning from Human Feedback
>
> [3] One-Shot Safety Alignment for Large Language Models via Optimal Dualization
>
> [4] Stepwise Alignment for Constrained Language Model Policy Optimization
>
> [5] Bounded Rationality for LLMs: Satisficing Alignment at Inference-Time
>
> [6] Rewards-in-Context: Multi-objective Alignment of Foundation Models with Dynamic Preference Adjustment

---

> > ### Comment · Reviewer_Qxpn · 2025-08-04
> > **Thank you for your response**
> >
> > Thank you for your thoughtful response and for conducting the additional experiment with multiple constraints. It has addressed my questions. I will increase the score.

---

> > > ### Author Response · Authors · 2025-08-04
> > >
> > > We are glad that your concerns have been addressed. We appreciate your thoughtful evaluation of our responses. Thank you very much for raising the score. We will incorporate the additional experiments and discussions into the final version of our paper. Thanks again.

---

### Official Review · Reviewer_VTYk · 2025-07-03

**Clarity:** 3
**Significance:** 3
**Originality:** 3
**Rating:** 5
**Confidence:** 2

**Summary:**

In this paper, the author proposes CAID, an alignment framework for language models that allows maximizing a primary reward objective while satisfying constraints on secondary objectives. The proposed method formulates this as a constrained optimization problem solved via duality theory. The approach is evaluated on PKU-SafeRLHF benchmark with respect to average reward score over baseline alignment methods such as DPO and shows a better balance between different alignment objectives.

**Questions:**

Refer Weaknesses

**Ethical Concerns:**

["NO or VERY MINOR ethics concerns only"]

**Final Justification:**

I went through the additional experiments and discussions in the rebuttal, and it addresses most of my concerns. Hence, for the final rating, I am recommending "Accept".

**Limitations:**

Yes

**Paper Formatting Concerns:**

No formatting issues

**Quality:**

3

**Strengths And Weaknesses:**

> **Strengths:**

1. The proposed approach appears sound and well-motivated. The claims are generally well-supported through theoretical analysis with bounded sub-optimality guarantees.

2. Empirical evaluation on safety alignment based on PKU-SafeRLHF validates the efficacy of the proposed approach.



> **Weaknesses:**

1. Evaluation has been done on only one dataset: PKU-SafeRLHF with respect to helpfulness, and harmlessness. The paper would benefit from evaluations on additional benchmarks (refer section 4 in [1]).

2. A recent work [2] also studied the problem of constraint satisfaction through controlled decoding. Further additional comparisons against multi-objective decoding approaches such as [1,3] would also help the paper.

3. Currently the paper lacks scalability analysis. It would be interesting to see how the performance of CAID varies as the number of reward dimensions increases.

**References:**

[1] Shi, R., Chen, Y., Hu, Y., Liu, A., Hajishirzi, H., Smith, N.A. and Du, S.S., 2024. Decoding-time language model alignment with multiple objectives. Advances in Neural Information Processing Systems, 37, pp.48875-48920.

[2] Chehade, M., Ghosal, S.S., Chakraborty, S., Reddy, A., Manocha, D., Zhu, H. and Bedi, A.S., 2025. Bounded Rationality for LLMs: Satisficing Alignment at Inference-Time. arXiv preprint arXiv:2505.23729.

[3] Chen, R., Zhang, X., Luo, M., Chai, W. and Liu, Z., 2024. Pad: Personalized alignment of llms at decoding-time. arXiv preprint arXiv:2410.04070.

---

> ### Author Rebuttal · Authors · 2025-07-31
>
> ## Re: Weakness 1
> Thank you for the suggestion. We have conducted additional experiments on a multi-constraint task involving two distinct constraints (harmlessness and humor) motivated by the Helpful Assistant task in [1]. Specifically, we fine-tune a LLaMA2-7B model on the Anthropic HH-RLHF dataset [4] to obtain a reference policy. We then align the model using our method under varying constraint levels. We use the helpfulness and harmlessness reward models from [6] for evaluating helpfulness and harmlessness, respectively, and the humor–no-humor model as the reward model for humor. We demonstrate that our iterative method satisfies the constraints while maximizing the primary utility.
>
> For implementation, similar to our main experiment, we randomly select 1000 prompts from the full HH-RLHF dataset and sample 128 responses per prompt in the dual sub-gradient step. In the policy optimization step, we train the model using DPO with 10K preference pairs randomly selected from the same dataset. We evaluate our trained models on a randomly sampled set of 2000 prompts, each with 2 responses, from the test split of the HH-RLHF dataset.
>
> | Criterion | Expected improvement | Actual improvement      |
> |-----------|----------------------|-------------------------|
> | Harmless  | 0.5                  | 0.588 (-0.765 → -0.177) |
> | Humor     | 0.5                  | 0.907 (-0.110 → 0.798)  |
>
> The table above shows the results for threshold $\mathbf{b}=[0.5,0.5]$, i.e., an expected improvement of 0.5 in both harmlessness and humor scores. We observe that both constraints are satisfied, while the helpfulness score improves by 9.12%.
>
> We are happy to report additional results for other values of $\mathbf{b}$ during the discussion period and will include this multi-constraint task in the final version of our paper.
>
> For additional benchmarks related to our main experiment, we have evaluated our method against the beaver-7b-v1.0 model from Safe-RLHF, as well as the best reported models, SACPO and P-SACPO, from [5]. For evaluation, we use the gpt-4o-mini model as the judge and the prompt dataset used in the GPT-4 evaluation from the Safe-RLHF paper. We observe that for constraint levels $\mathbf{b}=8,9$, our method in general achieves higher win rates in both helpfulness and safety criteria (see the table below).
> |                | **$\mathbf{b}=8$**     |        | **$\mathbf{b}=9$**     |        |
> |----------------|-------------|--------|-------------|--------|
> |                | Helpfulness | Safety | Helpfulness | Safety |
> | beaver-7b-v1.0 | 0.58        | 0.49   | 0.6         | 0.54   |
> | sacpo          | 0.62        | 0.53   | 0.78        | 0.56   |
> | p-sacpo        | 0.54        | 0.51   | 0.52        | 0.52   |
>
>
> ## Re: Weakness 2
> We would like to clarify two key differences between these papers and our work. First, we have different alignment objectives: the mentioned papers focus on exploring trade-offs among multiple objectives or properties, whereas our primary objective is to impose constraints on LLM models, i.e., to find an optimal *feasible* LLM model. It is well established that the Pareto optimality of multi-objective optimization is inherently different from optimality under constraints. Additionally, the multi-objective alignment papers typically do not suggest constrained satisfaction as we do in Figure 2(a).
>
> Second, while those works focus on inference-time alignment, our approach addresses offline alignment. Thus, our method and the referenced approaches are orthogonal and each applicable to different tasks and settings.
>
> Last but not least, our method is complementary to inference-time alignment. For instance, our iterative dualization approach can be directly applied to the constrained inference-time alignment to ensure constrained satisfaction, an extension we leave for future work.
>
> We appreciate the reviewer for bringing up these decoding-time approaches. We will incorporate the comparisons in the final version of the paper.
>
> ## Re: Weakness 3
> Our problem formulation and algorithm are not limited to single-constraint alignment and are readily scalable to multi-constraint settings. To demonstrate this, we will include the additional multi-constraint experiment in our response to Weakness 1 in the final version of our paper.
>
> We appreciate the reviewer's positive evaluation of our work, and believe that the additional experiments and discussions help address the concern regarding the evaluation of our method.
>
> **References:**
>
> [1]  Decoding-time language model alignment with multiple objectives.
>
> [2] Bounded Rationality for LLMs: Satisficing Alignment at Inference-Time.
>
> [3] Pad: Personalized alignment of llms at decoding-time.
>
> [4] Training a Helpful and Harmless Assistant with Reinforcement Learning from Human Feedback
>
> [5] Stepwise Alignment for Constrained Language Model Policy Optimization
>
> [6] Rewards-in-Context: Multi-objective Alignment of Foundation Models with Dynamic Preference Adjustment

---

### Official Review · Reviewer_GuGo · 2025-07-03

**Clarity:** 3
**Significance:** 2
**Originality:** 3
**Rating:** 4
**Confidence:** 3

**Summary:**

This paper presents an iterative dual-based method, CAID, for constrained LLM alignment. The core contribution is a theoretical analysis of teh optimality gap in the LLM parameter space, linking it to a "parametrization gap". The proposed "multi-shot" methd is shown empirically to outperform a one-shot baseline in satisfying constraints and achieving better reward/utility trade-offs.

**Questions:**

1. The parametrization gap $\nu$ is central to your theory but seems hard to measure. Any practical intuition on how to estimate it for a given model, or how sensitive the method is to model choice (e.g., 7B vs 70B)?

2. Figure 2a shows a systematic bias in the one-shot method. Is this primarily due to gradient estimation error, or is it a more fundamental problem with the single-step approach that your theory explains?

3. The iterative CAID method appears computationally heavy, requiring a full DPO run at each step (Alg 1). Can you comment on the computational cost vs. the one-shot method? How many iterations were typically needed to see the benefits shown in Fig. 2?

**Ethical Concerns:**

["NO or VERY MINOR ethics concerns only"]

**Final Justification:**

Thank you for addressing my concerns, I am raising the score from 3 to 4.

**Limitations:**

Yes

**Paper Formatting Concerns:**

Yes

**Quality:**

3

**Strengths And Weaknesses:**

**Strengths:**

*   **Rigorous Theoretical Analysis:** The key strength is the theoretical treatment. The paper moves beyond distribution-space analysis to provide optimality bounds (Thms 1-4) in the actual LLM parameter space. Formalizing the impact of the "parametrization gap" $\nu$ is a solid contribution that helps bridge theory and practice.
*   **Clear Problem Formulation:** The distinction between the distribution-space problem (U-CA) and the parameter-space problem (P-CA) is very well-articulated. This framing is crucial for understanding the core challenge and the paper's contribution.
*   **Well-Motivated Method:** The proposed CAID method is a natural and logical extension of prior one-shot work [19]. Applying an iterative dual ascent scheme is a principled way to address the limitations of a single-step optimization.
*   **Strong Empirical Demonstration:** While the scope is limited, the experiments are direct and effective. Figure 2 clearly shows the multi-shot approach better adheres to target constraints (Fig 2a) and achieves a superior Pareto frontier (Fig 2b) compared to the one-shot method. The results convincingly support the paper's claims.

**Weaknesses:**

*   **Limited Empirical Scope:** Experiments are confined to a single task (safety), one dataset (PKU-SafeRLHF), and 7B-scale models. The benefit of the iterative approach needs to be demonstrated on a wider range of constraints, tasks, and model scales to fully establish its general utility.
*   **Practicality of Assumptions:** The theory relies on several strong assumptions, like local strong convexity of the dual function (Assumption 5) and the existence of the parametrization gap $\nu$ (Assumption 3). The paper could benefit from more discussion on when these assumptions might hold or fail in real-world LLM training scenarios.
*   **Incremental Algorithmic Novelty:** The core algorithm (CAID) is an application of a standard dual sub-gradient method. The main novelty lies in its specific application to LLMs and the accompanying theoretical guarantees, not in the optimization algorithm itself.
*   **Abstractness of Parametrization Gap:** The parametrization gap $\nu$ is central to the theoretical bounds but remains an abstract quantity. The paper offers little guidance on how one might estimate $\nu$ or what architectural choices might minimize it, which limits the prescriptive power of the theory for practitioners.

---

> ### Author Rebuttal · Authors · 2025-07-31
>
> ## Re: Weekness 1
> Our problem formulation and algorithm are not limited to single-constraint alignment tasks and are readily extensible to multi-constraint settings. To address the reviewer’s concern, we have designed and conducted additional experiments on a multi-constraint task using a new dataset with two distinct constraints (harmlessness and humor). Specifically, we fine-tune a LLaMA2-7B model on the Anthropic HH-RLHF dataset [1] to obtain a reference policy. We then align the model using our method under varying constraint levels. We use the helpfulness and harmlessness reward models from [11] for evaluating helpfulness and harmlessness, respectively, and the humor–no-humor model as the reward model for humor. We demonstrate that our iterative method satisfies the constraints while maximizing the primary utility on this task.
>
> For implementation, similar to our main experiment, we randomly select 1000 prompts from the full HH-RLHF dataset and sample 128 responses per prompt in the dual sub-gradient step. In the policy optimization step, we train the model using DPO with 10K preference pairs randomly selected from the same dataset. We evaluate our trained models on a randomly sampled set of 2000 prompts, each with 2 responses, from the test split of the HH-RLHF dataset.
>
> | Criterion | Expected improvement | Actual improvement      |
> |-----------|----------------------|-------------------------|
> | Harmless  | 0.5                  | 0.588 (-0.765 → -0.177) |
> | Humor     | 0.5                  | 0.907 (-0.110 → 0.798)  |
>
> The table above shows the results for threshold $\mathbf{b}=[0.5,0.5]$, i.e., an expected improvement of 0.5 in both harmlessness and humor scores. We observe that both constraints are satisfied, while the helpfulness score improves by 9.12%.
>
> We are happy to report additional results for other values of $\mathbf{b}$ during the discussion period and will include this multi-constraint task in the final version of our paper.
>
> Regarding the concern about using the 7B-scale model, we note that our theoretical results and algorithm are applicable to LLMs of any size. The 7B-scale models are sufficient to address the questions we focus on, namely constraint satisfaction and trade-offs among properties (see Section 4.2). Moreover, using 7B models is a common practice in the alignment literature [2,3,4,5], and adhering to this scale facilitates fair and direct comparison with prior work.
>
>
> ## Re: Weakness 2
> The assumptions used in our analysis are intended to reflect realistic LLM training settings. First, Assumption 1 requires that the constraint functions be linearly independent over $m$ responses. This is easily satisfied in practice, as the response space of LLMs has a much higher dimensionality than the number of constraints $m$. Assumption 3 assumes that a generation distribution can be approximated by a practical LLM (i.e., a transformer network). Given the expressiveness of transformer (e.g., a universal approximation theory [6]), it is practical to assume the finiteness of the parametrization gap. We also note that KL divergence is widely used in regularized alignment problems to encourage the aligned model to remain close to a reference model. Similar assumptions have been adopted in prior work (e.g., [7]). In practice, the KL divergence can be evaluated to empirically verify the closeness between two LLMs (e.g., [1, 8]).
>
> In our optimization assumptions, we introduce a set of feasibility assumptions in LLM policy space (Assumption 2) and parameter space (Assumptions 4 and 6). The rationale of Assumptions 2 and 4 is the same as the feasibility in constrained optimization: a set of constraints can be satisfied by some points in the policy and parameter spaces. Empirically, we have evaluated the feasibility of aligned LLMs using different constraint thresholds (see Figure 2(a)), which is not provided by the previous constrained alignment studies. We use Assumption 6 for a technical reason as follows. It is known that the primal iterates induced by a dual-based method can only converge to an optimal one on average (e.g., see [9]), which is not desirable for LLMs. To show convergence of the primal iterates, the dual function has to be smooth, at least locally at some optimal dual variables. Thus, it is necessary to strengthen the feasibility with respect to optimal dual variables in Assumption 6. Last but not least, Assumption 5 assumes the local strong convexity of the dual function, which is indeed a consequence of Assumption 1 at an optimal dual variable. Empirically, it has been observed that the dual function is locally strongly convex (e.g., see Figure 1 of [3]). Therefore, the local strong convexity is a reasonable assumption in practice.
>
> We will incorporate these clarifications into the final version of our paper.
>
> ## Re: Weakness 3
> Among our key contributions, we show how to adapt and analyze the dual sub-gradient method in the context of LLMs, where the non-convexity of LLM parametrization presents a key challenge. Algorithmically, we instantiate the dual sub-gradient method as an iterative alignment algorithm, introducing warm-start initialization and post-processing (see Algorithm 1 and Appendix D) to improve convergence performance. A key algorithmic novelty is that we remove the evaluation of an optimal dual variable, which is often computed in a biased manner in practice (see [3]), and we introduce an iterative method to approximately find an optimal dual variable in an unbiased way. This technique generalizes the previous non-iterative method as a special case (e.g., [3]) and leads to significantly improved constraint satisfaction (see Figure 2).
>
> Theoretically, we characterize the duality gap of the constrained LLM alignment problem and quantify the optimality of the LLM policies induced by the dual iterations, which, to the best of our knowledge, have not been established in the existing literature. Our assumptions are tailored to the structure of LLM alignment, bridging the gap between constrained optimization and constrained alignment.
>
>
> ## Re: Weakness 4
> The primary goal of this work is to understand how the parameterization gap affects the performance of practically aligned LLMs, rather than to develop methods for minimizing it. Drawing from the universal approximation theory of transformers [6], we observe that there always exists a transformer (e.g., a transformer block with an attention layer with 2 heads of size 1 each, and a feed forward layer with 4 hidden nodes) capable of achieving a small parameterization gap. In theory, a zero parametrization gap is commonly assumed in the development of LLM alignment methods (e.g., DPO [10]), and this assumption has led to strong practical alignment performance. In practice, most LLMs (e.g., GPT-3 small: 12 layers with 12 heads of size 768 each, and a feedforward layer with 3072 hidden notes) are significantly overparametrized, which helps ensure that the parameterization gap remains small. Therefore, our theoretical results are prescriptive for aligning practical LLMs under constraints. Last but not least, the parametrization gap can be estimated in practice using the KL divergence estimates (e.g., [1, 8]) to compare an aligned model against a much larger model. We will add a discussion on this in the final version.
>
>
> ## Re: Question 1
> We note that practical LLMs are typically highly overparametrized, making it reasonable to expect that increasing model size generally reduces the parametrization gap due to improved expressiveness. To estimate the parametrization gap for a given model class, one can treat a much larger model class (e.g., 70B) as a proxy for the ideal model and estimate the KL divergences between the model and a few sampled proxy models (or a reference model) using standard estimation techniques in [1, 8]. Since the KL divergence serves as an upper bound on the 1-norm distance, it provides a useful estimate of the parametrization gap for a given model class. We will clarify these points in the final version.
>
>
> ## Re: Question 2
> The systematic bias observed in the one-shot empirical results can be attributed to the parametrization gap and optimality of the dual variables (see Theorem 4). In our multi-shot empirical results, we demonstrate that this bias can be significantly reduced by using instance-dependent dual variables (see Theorem 3). We will include a discussion of this point in the final version.
>
>
> ## Re: Question 3
> The computational cost of our iterative method is comparable to that of the one-shot method, as each iteration requires only a single epoch of DPO. Improvements shown in Fig. 2 are typically observed within four iterations. The additional runtime of the multi-shot method only comes from dual variable estimation, which can be efficiently accelerated through engineering techniques. Please refer to Appendix D.2 for detailed runtime comparisons.
>
> We appreciate the reviewer’s comments, and believe that the additional experiments and discussions provide further insight to open your evaluation of our work.
>
> **References:**
>
> [1] Training a Helpful and Harmless Assistant with Reinforcement Learning from Human Feedback
>
> [2] Safe RLHF: Safe Reinforcement Learning from Human Feedback
>
> [3] One-Shot Safety Alignment for Large Language Models via Optimal Dualization
>
> [4] Stepwise Alignment for Constrained Language Model Policy Optimization
>
> [5] Bounded Rationality for LLMs: Satisficing Alignment at Inference-Time
>
> [6] Are transformers universal approximators of sequence-to-sequence functions?
>
> [7] Theoretical Analysis of KL-regularized RLHF with Multiple Reference Models
>
> [8] Scaling Laws for Reward Model Overoptimization
>
> [9] Approximate Primal Solutions and Rate Analysis for Dual Subgradient Methods
>
> [10] Direct Preference Optimization: Your Language Model is Secretly a Reward Model
>
> [11] Rewards-in-Context: Multi-objective Alignment of Foundation Models with Dynamic Preference Adjustment

---

> ### Author Response · Authors · 2025-08-08
>
> Dear Reviewer GuGo, As the author-reviewer discussion window is nearing its end, we would greatly appreciate it if you could let us know whether our updates address your concerns. We have included an additional experiment on a different task with multiple constraints, a new dataset, and a new model to further demonstrate the empirical scope of our work, and we have responded to all of your questions. We are happy to engage in further discussion or provide additional clarification if you have any remaining questions.

---

### Author Response · Authors · 2025-08-09
**Summary of Rebuttal**

As the discussion period draws to a close, we would like to sincerely thank all reviewers for their thoughtful feedback and constructive suggestions on our paper. We appreciate the reviewers’ recognition of our paper’s strengths, including its rigorous theoretical analysis, clear problem formulation, and principled multi-shot method. The reviewers also recognized our strong empirical results on constraint satisfaction and the trade-offs between helpfulness and safety, as well as the clear presentation with broader societal benefits.

To address the main concern regarding the empirical scope of our experiments beyond the single-constraint task, we have conducted additional experiments on a multi-constraint problem using a new model and a new dataset. The results demonstrate our algorithm’s effectiveness in a different experimental setting. We also have conducted additional comparisons to broader benchmarks as suggested, further strengthening our empirical evaluation.

In addition, we have provided detailed responses to each reviewer’s questions. As of this writing, three reviewers who have responded to our rebuttal indicated that their concerns have been satisfactorily addressed.

For the final version of our paper, we plan to:
- Include the multi-constraint experiments with varying constraints and corresponding visualizations
- Add the suggested benchmark citations and comparisons
- Provide further clarification on the practicality of the assumptions, as well as the novelty of the algorithm and analysis

We believe our work makes a significant contribution to the constrained LLM alignment problem by characterizing the duality gap and establishing the optimality of the LLM policies induced by the dual iterations. Furthermore, our proposed algorithm effectively ensures constraint satisfaction and significantly improves trade-off between constraints.

---

### Decision · Program_Chairs · 2025-09-17

**Decision:**

Accept (poster)

**Comment:**

This paper presents a principled approach to constrained LLM alignment through the CAID (Constrained Alignment via Iterative Dualization) method, which extends prior one-shot approaches to a multi-shot framework. The theoretical contributions are solid, including characterizing primal-dual gaps, quantifying optimality with respect to both objectives and constraints. During the reviewing process, reviewers initially raised concerns about the limited empirical scope (single constraint, 7B models), the authors addressed these through additional multi-constraint experiments on the Anthropic HH-RLHF dataset and comparisons with broader baselines. Furthermore, they provided clarifications regarding the practicality of their theoretical assumptions and the novelty of their approach in the non-convex LLM setting. The resulting paper is a solid contribution that advances the theoretical understanding and practical application of constrained LLM alignment.